# Assessing the efficacy, safety and utility of 6-month day-and-night automated closed-loop insulin delivery under free-living conditions compared with insulin pump therapy in children and adolescents with type 1 diabetes: an open-label, multicentre, multinational, single-period, randomised, parallel group study protocol

Gianluca Musolino,[1,2] Janet M Allen,[1,2] Sara Hartnell,[3] Malgorzata E Wilinska,[1,2] Martin Tauschmann,[1,4] Charlotte Boughton,[1] Fiona Campbell,[5] Louise Denvir,[6] Nicola Trevelyan,[7] Paul Wadwa,[8] Linda DiMeglio,[9] Bruce A Buckingham,[10] Stuart Weinzimer,[11] Carlo L Acerini,[1,2] Korey Hood,[10] Steven Fox,[12] Craig Kollman,[13] Judy Sibayan,[13] Sarah Borgman,[13] Peiyao Cheng,[13] Roman Hovorka[1,2]

For numbered affiliations see end of article.

**Correspondence to**
Professor Roman Hovorka;
rh347@cam.ac.uk

## ABSTRACT

**Introduction** Closed-loop systems titrate insulin based on sensor glucose levels, providing novel means to reduce the risk of hypoglycaemia while improving glycaemic control. We will assess effectiveness of 6-month day-and-night closed-loop insulin delivery compared with usual care (conventional or sensor-augmented pump therapy) in children and adolescents with type 1 diabetes.

**Methods and analysis** The trial adopts an open-label, multicentre, multinational (UK and USA), randomised, single-period, parallel design. Participants (n=130) are children and adolescents (aged ≥6 and <19 years) with type 1 diabetes for at least 1 year, and insulin pump use for at least 3 months with suboptimal glycaemic control (glycated haemoglobin ≥58 mmol/mol (7.5%) and ≤86 mmol/mol (10%)). After a 2–3 week run-in period, participants will be randomised to 6-month use of hybrid closed-loop insulin delivery, or to usual care. Analyses will be conducted on an intention-to-treat basis. The primary outcome is glycated haemoglobin at 6 months. Other key endpoints include time in the target glucose range (3.9–10 mmol/L, 70–180 mg/dL), mean sensor glucose and time spent above and below target. Secondary outcomes include SD and coefficient of variation of sensor glucose levels, time with sensor glucose levels <3.5 mmol/L (63 mg/dL) and <3.0 mmol/L (54 mg/dL), area under the curve of glucose <3.5 mmol/L (63 mg/dL), time with glucose levels >16.7 mmol/L (300 mg/dL), area under the curve of glucose >10.0 mmol/L (180 mg/dL), total, basal and bolus insulin dose, body mass index z-score and blood pressure. Cognitive, emotional and behavioural characteristics of participants and caregivers and their responses to the closed-loop and clinical trial will be assessed. An incremental cost-effectiveness ratio for closed-loop will be estimated.

**Ethics and dissemination** Cambridge South Research Ethics Committee and Jaeb Center for Health Research Institutional Review Office approved the study. The

## Strengths and limitations of this study

► The study adopts an open-label, multicentre, multinational, randomised, parallel design: it includes a large group of children and adolescents across wide geographical locations.

► The trial adopts a 6-month follow-up period of hybrid closed-loop insulin delivery during unrestricted living.

► Participants in the two study groups will have an equal number of study visits.

► The study design excludes participants with recurrent incidents of severe hypoglycaemia or diabetic ketoacidosis during the previous 6 months, living alone and those with glycated haemoglobin <58 mmol/mol (7.5%) and >86 mmol/mol (10%) and with high or very low daily insulin requirements (total daily insulin dose ≥2 IU/kg/day or <15 IU/day).

► All participants are already pump users, somewhat limiting generalisability.

findings will be disseminated by peer-review publications and conference presentations.

**Trial registration number** NCT02925299; Pre-results.

## INTRODUCTION

Type 1 diabetes is characterised by a deficiency of insulin caused by immunologically mediated damage to pancreatic beta cells, leading to raised blood glucose levels. Diabetes is one of the most common metabolic conditions. It is estimated that in 2017, 1 100 000 children and adolescents (0–19 years) worldwide had type 1 diabetes and that the number of newly diagnosed cases was over 130 000.[1] The incidence rate in children is increasing by ~3%–4% per year with geographic differences.[1] Earlier onset can result in diabetes complications appearing at a younger age, while dependence on lifelong insulin imposes a heavy burden on children, carers as well as healthcare systems.

Despite continuing progress, glycaemic control in children and adolescents with type 1 diabetes remains suboptimal.[2] The achievement of recommended treatment goals is limited by the risk of hypoglycaemia. Even in those with the desired level of glycaemic control, non-physiological glucose excursions occur with periods of silent hyperglycaemia and hypoglycaemia.[3 4] Individuals have blunted counter-regulatory responses to hypoglycaemia impairing recovery and increasing the threat of future episodes.[5] Recurrent episodes may lead to hypoglycaemic unawareness, increasing the risk of severe hypoglycaemia.[6] Hypoglycaemia has psychological consequences including the fear of hypoglycaemia with resulting maladaptive coping behaviours, such as excessive eating or under-insulinising, that may negatively impact glycaemic control.[7]

The development of continuous glucose monitoring has been a major advance.[8–11] Sensor-augmented pumps combine real-time continuous glucose monitoring with insulin pump.[12] Insulin pumps with low-glucose suspend feature have been shown to reduce hypoglycaemia.[13] These systems, however, overall provide little or no automation to adjust insulin delivery to match glucose excursions.

An artificial pancreas (a closed-loop system) adjusts insulin automatically and represents a realistic treatment option for type 1 diabetes.[14] The closed-loop control algorithm translates, in real-time, sensor glucose levels received from the glucose monitoring device and computes the amount of insulin to be delivered by the coupled insulin pump. Hybrid closed-loop systems automatically titrate insulin delivery, although the user manages insulin boosts at meal time.[15] In 2017, the first closed-loop system entered clinical use in the USA.[16]

Closed-loop systems may improve glycaemic control while reducing the risk of hypoglycaemia.[17] They have been evaluated in children and adolescents under controlled laboratory conditions[18–20] and in home settings.[21–24] Investigations in adults have also been conducted.[22 25 26] Psychosocial assessments support

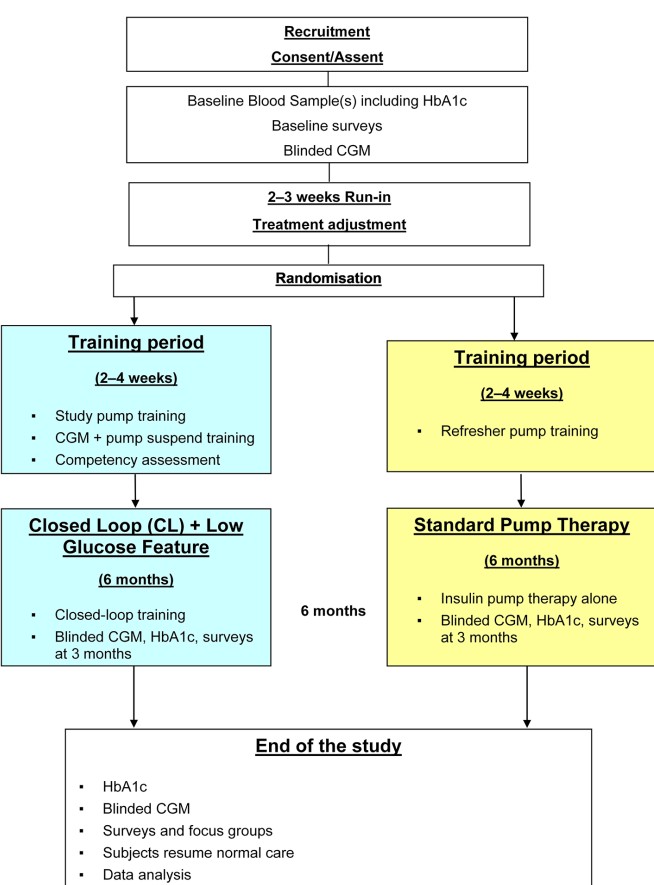

**Figure 1** Study flowchart. HbA1c, glycated haemoglobin; CGM, continuous glucose monitoring.

acceptability and benefits of this therapeutic approach among children/adolescents and carers.[27] Closed-loop systems are associated with increased time in near normoglycaemia and reduced time in hypoglycaemia and hyperglycaemia.[28] So far, evaluations have been limited to 3 months.[22]

The present study will assess the efficacy, safety, utility and acceptability of 6-month day-and-night hybrid closed-loop insulin delivery during unrestricted living in comparison to usual care in children and adolescents with type 1 diabetes.

## METHODS AND ANALYSIS
### Overview

This trial adopts an open-label, multicentre, multinational, single-period, randomised, parallel group design, involving a 6-month home study period during which day-and-night glucose levels will be managed either by a closed-loop system (intervention group) or by insulin pump therapy (control group) (figure 1). We aim to recruit up to 150 children and adolescents aged ≥6 to <19 years with type 1 diabetes on insulin pump therapy (approximately equal proportion of those aged ≥6 to 12 years and 13 to <19 years, a minimum quota of 25% participants with baseline glycated haemoglobin

>69 mmol/mol, >8.5%). Inclusion and exclusion criteria are summarised in box 1.

The University of Cambridge (UK) and Jaeb Center for Health Research (USA) are the coordinating centres. Clinical centres include:

1. Addenbrooke's Hospital, Cambridge, UK.
2. Barbara Davis Center for Childhood Diabetes, Aurora, USA.
3. Indiana University, Indianapolis, USA.
4. Leeds Teaching Hospital, Leeds, UK.
5. Nottingham Children's Hospital, Nottingham, UK.
6. Southampton Children's Hospital, Southampton, UK.
7. Stanford University, Stanford, California, USA.
8. Yale University, New Haven, Connecticut, USA.

Cognitive, emotional and behavioural characteristics of participants and family members and their response to the closed-loop will be assessed gathering both quantitative (validated surveys) and qualitative data (focus groups). Written informed consent/assent will be obtained from all participants and guardians before any study-related activities.

### Study schedule

The study will comprise up to eight visits and six telephone/email contacts (see tables 1 and 2). The maximum study duration is 8 months.

### Screening and baseline assessment

At screening, blood samples for full blood count, liver, thyroid function and anti-transglutaminase antibodies (with IgA levels if not done within previous 12 months) will be taken. Non-hypoglycaemia C-peptide, glucose and glycated haemoglobin will be measured and a urine pregnancy test in females of childbearing potential will be performed. Surveys investigating participants' quality of life, psychosocial and cognitive functioning, and response to their current treatment will be distributed. Participants will be fitted with a blinded continuous glucose monitoring device (Libre Pro, Abbott Diabetes Care, Alameda, CA, USA) that will be worn during the run-in period at home for up to 14 days.

### Run-in period

During a 2–3 week run-in period, subjects will continue using their own insulin pump. Data obtained from blinded glucose sensors and pump downloads may be utilised for treatment adjustments. The run-in period may be extended/repeated if no or limited sensor data are available. At least 10 days of sensor data need to be collected. A longer run-in will not be used for additional fine-tuning of treatment adjustments.

### Randomisation

Central randomisation software will be used with stratification by site and baseline glycated haemoglobin. The randomisation ratio will be 1:1 within each stratum. The randomisation list created by the study statistician is encrypted.

---

**Box 1  Inclusion and exclusion criteria**

Summary of inclusion criteria
- Age ≥6 and <19 years.
- Type 1 diabetes as defined by WHO[34] for at least 1 year.
- Use of an insulin pump for at least 3 months, with good knowledge of insulin self-adjustment by subject or caregiver as judged by the investigator.
- Using U-100 rapid acting insulin analogues Aspart or Lispro only.
- Willing to perform regular finger-prick blood glucose monitoring, with at least 4 blood glucose measurements per day.
- Screening glycated haemoglobin ≥58 mmol/mol (7.5%) and ≤86 mmol/mol (10%) based on analysis from local laboratory.
- Literate in English.
- Willing to wear continuous glucose sensor and closed-loop system at home.
- Willing to follow study-specific instructions.
- Willing to upload pump and glucose sensor data at regular intervals.
- Access to Wi-Fi.
- Living with someone who is trained to administer glucagon and is able to seek emergency assistance.

Summary of exclusion criteria
- Living alone.
- Current use of any closed-loop system.
- Any other physical or psychological disease likely to interfere with the normal conduct of the study and interpretation of the study results, as judged by the investigator.
- Untreated coeliac disease, adrenal insufficiency or untreated thyroid disease.
- Current treatment with drugs known to interfere with glucose metabolism (eg, systemic corticosteroids, non-selective beta-blockers, monoamine oxidase inhibitors and so on).
- Known or suspected allergy to insulin.
- Clinically significant nephropathy (estimated glomerular filtration rate <45 mL/min) or on dialysis, neuropathy or active retinopathy (presence of maculopathy or proliferative changes), as judged by the investigator.
- Recurrent incidents of severe hypoglycaemia (>1 episode) during the previous 6 months (adolescents: severe hypoglycaemia is defined as an event requiring assistance of another person to actively administer carbohydrates, glucagon or take other corrective actions including episodes of hypoglycaemia severe enough to cause unconsciousness, seizures or attendance at hospital; children: severe hypoglycaemia is defined as an event associated with a seizure or loss of consciousness).
- Recurrent incidents of diabetic ketoacidosis (>1 episode) during the previous 6 months.
- Unwilling to avoid regular use of acetaminophen.
- Lack of reliable telephone facility for contact.
- Total daily insulin dose ≥2 IU/kg/day and <15 IU/day.
- Pregnancy, planned pregnancy or breast feeding.
- Severe visual or hearing impairment.
- Seizure disorder.
- Medically documented allergy towards the adhesive (glue) of plasters or unable to tolerate tape adhesive in the area of sensor placement.
- Serious skin diseases (eg, psoriasis vulgaris, bacterial skin diseases) located at places of the body likely to be used for localisation of the glucose sensor.
- Abusing illicit drugs, prescription drugs or alcohol.

Continued

---

## Box 1   Continued

► Use of pramlintide (Symlin), or other non-insulin glucose lowering agents including sulphonylureas, biguanides, dipeptidyl peptidase-4 (DPP4) inhibitors, glucagon-like peptide-1 (GLP-1) analogues, sodium-glucose cotransporter (SGLT)-1/2 inhibitors at time of screening.
► Shift work with working hours between 22:00 and 08:00.
► Sickle cell disease, haemoglobinopathy or has received red blood cell transfusion or erythropoietin within 3 months prior to time of screening.
► Eating disorder such as anorexia or bulimia.
► Employed by Medtronic Diabetes or with immediate family members employed by Medtronic Diabetes.

### Treatment period

#### Automated day-and-night hybrid closed-loop insulin delivery combined with low-glucose suspend feature (interventional arm)

Participants allocated to the closed-loop group will be trained on using the study insulin pump (modified Medtronic 640G pump, Medtronic, Northridge, CA, USA) and real-time continuous glucose sensor (Guardian 3, Medtronic). This represents a complex intervention over usual care, especially for subjects under pump therapy alone. Once deemed competent with the use of the devices, participants will receive training required for the closed-loop system. Competency on the use of closed-loop will be evaluated. During closed-loop period, participants will programme meal boluses estimating ingested carbohydrate amounts. Specific instructions during closed-loop related to exercise management, sick day rules, hypoglycaemia and hyperglycaemia management and technical troubleshooting will be provided.

#### Usual care (conventional or sensor-augmented pump therapy) (control arm)

Participants in control arm will receive refresher training on key aspects of insulin pump therapy (advanced boluses, temporary basal, infusion set change, sensor calibrations). During 6-month control intervention period, subjects will continue using either their own insulin pump alone or combined with their prestudy glucose monitoring device.

**Table 1**  Schedule of study visits/phone contacts when the participant is randomised to day-and-night closed-loop combined with low-glucose feature (intervention group)

|  | Visit/contact | Description | Start relative to previous/next visit/activity | Duration (hours) |
|---|---|---|---|---|
| Run-in | Visit 1 | Recruitment visit: consent, HbA1c, screening bloods, urine pregnancy test, baseline surveys, blinded CGM training and insertion |  | 1–4 |
|  | Visit 2 | Review of baseline bloods, pump settings and CGM data; adjustment of treatment | 2 weeks after visit 1 (+1 week); run-in could be repeated | 1–2 |
| Training period | Visit 3 | Randomisation, repeat HbA1c if visit 3 and visit 1 are >28 days apart, urine pregnancy test, study pump training and initiation, competency assessment | May coincide with visit 2, within 8 weeks of visit 1 | 3–4 |
|  | Visit 3a | Real-time CGM training and initiation, competency assessment | Within 0–7 days of visit 3 (visit 3a may coincide with visit 3; training visits can be repeated) | 2–4 |
| CL + LGS intervention (6 months) | Visit 4* | CL initiation at clinic/home: data download, CL and low-glucose feature training, competency assessment, blinded CGM | 4 weeks after randomisation (±1 week) | 2–6 |
|  | Contact 1 | Review use of study devices; study update | Within 24–48 hours after visit 4 | <1 |
|  | Visit 5† | Review use of study devices; study update | 1 week after visit 4 (±3 days) | <1 |
|  | Contact 2 | Review use of study devices; study update | 2 weeks after visit 4 (±3 days) | <1 |
|  | Contact 3 | Review use of study devices; study update | 1 month after visit 4 (±2 weeks) | <1 |
|  | Contact 4 | Review use of study devices; study update | 2 months after visit 4 (±2 weeks) | <1 |
|  | Visit 6 | 3 month visit: HbA1c, urine pregnancy test, data download, blinded CGM, surveys | 4 months after randomisation (±2 weeks) | 1–3 |
|  | Contact 5 | Review use of study devices; study update | 5 months after randomisation (±2 weeks) | <1 |
|  | Contact 6 | Review use of study devices; study update | 6 months after randomisation (±2 weeks) | <1 |
|  | Visit 7 | Blinded CGM | 2–4 weeks before planned visit 8 | <0.5 |
|  | Visit 8 | End of closed-loop treatment arm (6 months of CL): HbA1c, data download, surveys and focus groups; resume usual pump therapy | 7 months after randomisation (±2 weeks) | 1–3 |

*In-person clinic visit mandatory in USA only.
†Could be done via phone/email in UK. In-person visit mandatory in USA only.
CGM, continuous glucose monitoring; CL, closed-loop; HbA1c, glycated haemoglobin; LGS, low-glucose suspend.

**Table 2** Schedule of study visits/phone contacts when the participant is randomised to usual care (conventional or sensor-augmented insulin pump therapy) (control group)

| | Visit/contact | Description | Start relative to previous/next visit/activity | Duration (hours) |
|---|---|---|---|---|
| Run-in | Visit 1 | Recruitment visit: consent, HbA1c, screening bloods, urine pregnancy test, baseline surveys, blinded CGM training and insertion | | 1–4 |
| | Visit 2 | Review of baseline bloods, pump settings and CGM data; adjustment of treatment | 2 weeks after visit 1 (+1 week); run-in could be repeated | 1–2 |
| Training period | Visit 3 | Randomisation, repeat HbA1c if visit 3 and visit 1 are >28 days apart, urine pregnancy test, insulin pump refresher training, competency assessment | May coincide with visit 2, within 8 weeks of visit 1 | 3–4 |
| Usual insulin pump therapy intervention (6 months) | Visit 4* | Initiation of standard therapy arm at clinic/home, glucometer download, recording of current insulin requirements, blinded CGM | 4 weeks after randomisation (±1 week) | 2–6 |
| | Contact 1 | Study update | Within 24–48 hours after visit 4 | <1 |
| | Visit 5† | Study update | 1 week after visit 4 (±3 days) | <1 |
| | Contact 2 | Study update | 2 weeks after visit 4 (±3 days) | <1 |
| | Contact 3 | Study update | 1 month after visit 4 (±2 weeks) | <1 |
| | Contact 4 | Study update | 2 months after visit 4 (±2 weeks) | <1 |
| | Visit 6 | 3-month visit: HbA1c, urine pregnancy test, glucometer download, recording of current insulin requirements, surveys, blinded CGM | 4 months after randomisation (±2 weeks) | 1–3 |
| | Contact 5 | Study update | 5 months after randomisation (±2 weeks) | <1 |
| | Contact 6 | Study update | 6 months after randomisation (±2 weeks) | <1 |
| | Visit 7 | Blinded CGM | 2–4 weeks before planned visit 8 | <0.5 |
| | Visit 8 | End of standard pump therapy treatment arm (6 months): HbA1c, glucometer download, recording of current insulin requirements, surveys and focus groups, resume usual care | 7 months after randomisation (±2 weeks) | 1–3 |

*In-person clinic visit mandatory in USA only.
†Could be done via phone/email.
CGM, continuous glucose monitoring; HbA1c, glycated haemoglobin.

At the study initiation visit, participants in both study groups will be fitted with a blinded continuous glucose monitoring system (Libre Pro) that will be worn for up to 14 days. If the sensor fails or gets detached, another sensor may be inserted. The sensor data may be used to optimise insulin delivery.

### Assessments at 3 and 6 months
A blood sample will be collected for measurement of glycated haemoglobin. A urine pregnancy test in females of childbearing potential will be performed. As per usual clinical practice, glucometer downloads and pump data will be reviewed, and adjustments to insulin pump settings will be made as required. Validated surveys evaluating the impact of the devices employed on quality of life, psychosocial and cognitive functioning, diabetes management and treatment satisfaction will be administered. At the 3-month follow-up visit, participants in both study groups will be fitted with blinded continuous glucose monitoring systems (Libre Pro). For assessment of glycaemic control during the final 3-month period of the trial, participants in both study groups will be fitted with a blinded

continuous glucose monitoring system 2–4 weeks before the end of study. At the 6-month visit, the same procedures as at the 3-month visit will be followed. A subset of subjects/guardians will be invited to join follow-up focus groups.

### Study contacts during 6-month study period
Participants in the two study groups will have an equal number of contact visits. The first planned contact will occur within 24–48 hours after study initiation visit. During the first 2 weeks of the study period, participants will be contacted weekly. Thereafter, participants will be contacted monthly. Subjects/parents and/or the clinical team are free to adjust insulin therapy, but no active treatment optimisation will be undertaken by the research team.

### Devices download
As per usual care, insulin pump and blood glucose meter will be downloaded (Medtronic CareLink) every clinic visit (at least every 3 months).

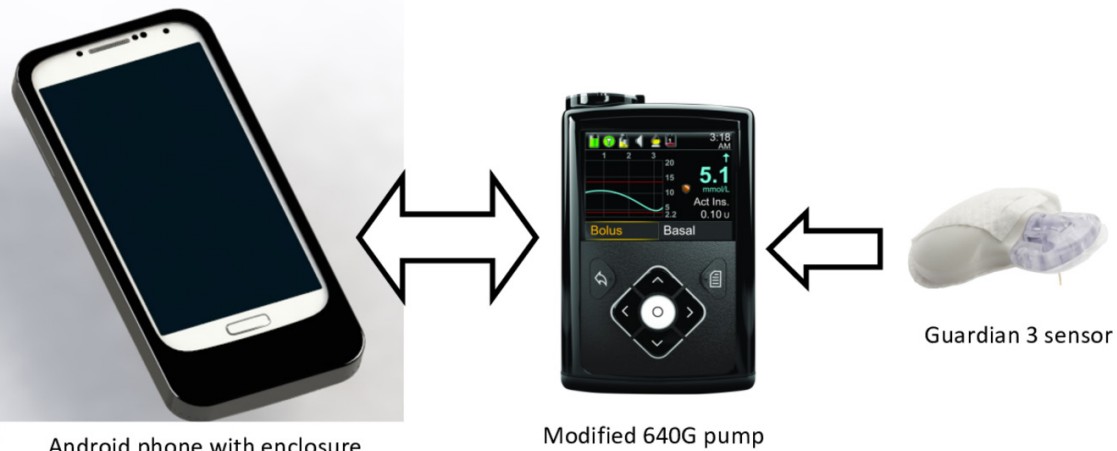

**Figure 2** FlorenceM closed-loop system prototype. The system consists of a continuous glucose monitoring transmitter with Guardian 3 sensor (Medtronic), an insulin pump (modified 640G pump, Medtronic) and an Android smartphone running the control algorithm (Cambridge).

## Closed-loop system

The FlorenceM closed-loop system (figure 2) incorporates a computer-based algorithm hosted by an Android smartphone, which interacts wirelessly with the modified investigational-use-only 640G pump through a proprietary translator device included in the smartphone's enclosure. By using the information received from the glucose sensor, every 10 min the system computes a new temporary basal insulin infusion rate, which is automatically sent to the insulin pump. The treat-to-target control algorithm aims to achieve a default glucose level of 5.8 mmol/L (104 mg/dL) and regulates the actual level depending on fasting versus postprandial status and the accuracy of model-based glucose predictions. No remote monitoring is planned. While the system is charging and connected to internet, the device uploads data on a server. The study pump comprises continuous glucose monitoring receiver and provides hypoglycaemia and hyperglycaemia alarms, which can be activated/personalised by the participants.

## Safety precautions during closed-loop

Participants will be asked to perform capillary calibrations before breakfast and dinner. If sensor glucose value is >3.0 mmol/L (54 mg/dL) different from capillary glucose level, the sensor will be recalibrated. These directions are based on an in silico simulation of hypoglycaemia and hyperglycaemia risk using the validated Cambridge simulator.[29] If sensor glucose becomes unavailable or the smartphone is not in range/operational, the pump will automatically deliver the preprogrammed insulin as set on the pump within 30 min. Safety rules limit maximum insulin infusion and suspend insulin delivery when sensor glucose is ≤4.3 mmol/L (77 mg/dL) or when glucose is rapidly decreasing. In case of a communication failure between control algorithm device and the study pump, the low-glucose feature will interrupt insulin delivery, provided sensor glucose is available. Low-glucose suspend/predictive low-glucose management will be

initially set to suspend insulin delivery at sensor glucose values of 3.9 mmol/L (70 mg/dL) or less, after which the setting could range from 2.8 to 5.0 mmol/L (50–90 mg/dL). Predictive low-glucose suspend will not be used. Insulin delivery will be resumed in accordance of the low-glucose suspend feature implemented on the study pump. A 24-hour local telephone helpline will be available for any technical device issues or problems related to diabetes management.

## Participant withdrawal criteria

The following prerandomisation withdrawal criteria will apply:
1. Subject/caregiver is unable to demonstrate safe use of study insulin pump as judged by the investigator.
2. Subject/caregiver fails to demonstrate compliance with insulin pump and capillary self-monitoring of blood glucose during run-in.

Prerandomisation and postrandomisation withdrawal criteria will comprise:
1. Subjects/caregivers may terminate participation in the study at any time without necessarily giving a reason and without any personal disadvantage.
2. Significant protocol violation or non-compliance.
3. Two distinct episodes of severe hypoglycaemia.
4. Two distinct episodes of diabetic ketoacidosis unrelated to infusion site failure and related to the use of the closed-loop.
5. Decision by the investigator or the sponsor that termination is in the subject's best medical interest.
6. Allergic reaction to insulin.
7. Allergic reaction to adhesive surface of infusion set or glucose sensor.
8. Subject becomes pregnant during the study period.

Subjects withdrawn due to reasons 4–10 will be invited to provide blood sample at the end of the planned study intervention for the assessment of glycated haemoglobin.

## Psychosocial evaluations

Cognitive, emotional and behavioural characteristics of participating subjects and family members and their response to the closed-loop system and clinical trial will be assessed using validated surveys and focus groups. Surveys will be completed at baseline (prior to randomisation), at 3 and 6 months.

To assess how strongly participants value the benefits of the closed-loop (compared with the usual care), we will conduct a discrete choice experiment (DCE). In the DCE, respondents will answer a series of binary choice questions (eg, 'Given a choice between option A or B, which would you prefer…') where those two options offer differing strengths and weaknesses. By varying the performance levels of these different desirable characteristics, we can assess their relative importance.

Focus groups will be completed at the end of the study (6 months). We will conduct virtual focus groups using HIPAA-approved software supported by Stanford University. Focus groups will be run with three to six participants and we will work from a script of open-ended questions used to gather feedback and reactions to the closed-loop system/insulin pump therapy, the clinical trial and quality of life changes. Sessions will be audio taped and video taped and transcribed by a professional transcription service.

## Blood samples

Screening blood samples will be measured locally. Additional blood samples will be taken for the measurement of non-hypoglycaemia C-peptide and glycated haemoglobin at a central laboratory. Glycated haemoglobin will be assessed at baseline, 3 and 6 months. At each time point, glycated haemoglobin will be measured locally (clinical care) and centrally (analysis of study endpoints). The central analysis will be performed using an International Federation of Clinical Chemistry and Laboratory Medicine aligned method.

## Patient and public involvement

The research question and study endpoints are based on feedback from participants of previous studies and in line with prioritising by stakeholders.[30] The study design and the assessment of the burden of the intervention were reviewed by focus groups. Results will be disseminated to participants and general public through social media and will be made available on the sponsor's website.

## Statistical analysis
### Primary outcome analysis

The primary analysis will follow the intention-to-treat principle. Data from all randomised subjects will be analysed in the group to which the subjects were assigned through randomisation regardless of the actual treatment received. Data will not be truncated due to protocol deviations.

The primary analysis will evaluate between group differences in glycated haemoglobin levels at the end of treatment period. A 5% significance level will be considered statistically significant for the primary outcome comparison.

Mean±SD values or percentiles appropriate to the distribution will be reported for the primary outcome by treatment group. The two treatment groups will be compared using a linear regression model adjusting for glycated haemoglobin at baseline, age and clinical centre as random effect. A 95% CI will be reported for the difference between the randomisation groups based on the linear regression model. Residual values will be examined for an approximate normal distribution. If values are highly skewed, then a transformation or robust statistical methods (eg, non-parametric or MM estimation) will be used instead. A detailed analysis plan will be provided separately.

### Other key endpoints

For the following key endpoints at 6 months, the family-wise type I error rate will be controlled at two-sided α=0.05. A gatekeeping strategy will be used, where the primary endpoint will be tested first, if passing the significance testing, other key endpoints will be tested in the order listed below using the fixed-sequence method at α=0.05.

► Time spent in the target glucose range from 3.9 to 10.0 mmol/L (70–180 mg/dL).
► Mean sensor glucose.
► Time spent above target glucose 10.0 mmol/L (180 mg/dL).
► Time spent below target glucose 3.9 mmol/L (70 mg/dL).

If a non-significant (p>0.05) result is obtained for any outcome on this list, no further hypothesis testing will be performed for any metrics further down on the list.

### Secondary efficacy analyses

For these exploratory analyses, the false discovery rate will be used to account for multiple comparisons:

#### Continuous glucose monitoring derived indices

► SD of sensor glucose.
► Sensor glucose variability measured with the coefficient of variation.
► The time with glucose <3.5 mmol/L (63 mg/dL).
► The time with glucose <3.0 mmol/L (54 mg/dL).
► Area under the curve of glucose <3.5 mmol/L (63 mg/dL).
► The time spent in significant hyperglycaemia (glucose >16.7 mmol/L, 300 mg/dL).
► Area under the curve of glucose >10.0 mmol/L (180 mg/dL).

The following sensor glucose metrics will also be calculated separately for day-time period (06:00–23:59) and night-time period (00:00–05:59):

► The time with glucose from 3.9 to 10.0 mmol/L (70–180 mg/dL).
► Mean glucose.
► Glucose variability as measured by SD.
► The time with glucose <3.5 mmol/L (63 mg/dL).

*Binary metrics for glycated haemoglobin*
► HbA1c <53 mmol/mol (7.0%).
► HbA1c <58 mmol/mol (7.5%).
► Relative reduction ≥10% from baseline.
► Absolute reduction ≥0.5% from baseline.
► Absolute reduction ≥1% from baseline.
► Absolute reduction ≥1% from baseline or HbA1c<53 mmol/mol (7.0%).

*Insulin and other endpoints*
► Total, basal and bolus insulin dose.
► Body weight (body mass index z-score).
► Blood pressure.

The above described glycaemic metrics will be based on sensor glucose levels collected during postrandomisation periods of blinded sensors wear.

### Safety analyses
The following events will be recorded and compared between treatment groups:
► Number of severe hypoglycaemia events per subject and incidence rate per 100 person-years.
► Number of diabetic ketoacidosis events per subject and incidence rate per 100 person-years.
► Sensor glucose-measured hypoglycaemic events per week (>15 min with glucose <3 mmol/L, 54 mg/dL).
► Sensor glucose-measured hyperglycaemic events per week (>15 min with glucose >16.7 mmol/L, 300 mg/dL).
► Proportion of subjects with worsening of glycated haemoglobin from baseline to 6 months by >0.5%.

If we record enough observed events to allow formal statistical modelling for above safety outcomes, we will perform the following analyses. Poisson regression models will be constructed to compare the treatment group difference for event rates by adjusting for age, baseline glycated haemoglobin and random site effect. If any outlier exists, a robust Poisson regression model will be used instead. For binary glycated haemoglobin outcome, logistic regression models will be used to compare the treatment group difference by adjusting for age, baseline glycated haemoglobin and random site effect.

### Utility assessments
The following system use/function outcomes in the intervention arm will be tabulated:
► Number of low-glucose suspend events.
► Percentage of time when closed-loop system use is functioning.
► Percentage of time when continuous glucose monitoring is used.

### Subgroup analyses
No subgroups were considered during the power calculations. Interpretation of any subgroup analyses will depend on whether the overall analysis demonstrates a significant treatment group difference. In the absence of such difference, if performed, the subgroup analyses will be interpreted with caution.

### Psychosocial analyses
Quantitative data on usability and satisfaction will be analysed using simple descriptive statistics. Additionally, we will analyse scores from the cognitive, emotional and behavioural assessments to determine if changes occur over time and between groups.

We will construct predictive models in the general linear framework to examine the associations with primary outcomes. For the DCE, the strength of preference (importance) of each performance attribute will be estimated from the pooled DCE responses using standard regression analysis techniques.

Qualitative data will be analysed using Atlas.ti V.6.0 to organise and manage the entire corpus of focus group data.

### Cost utility analyses
To inform reimbursement and other policy decision-making, we will conduct a cost utility analysis on the benefits of closed-loop. The analysis timeframe for both costs and benefits will include not just the study period but also anticipated future impacts. Costs will be denominated in US$. They will be framed to include both health-related expenditures and any realised or projected incremental health cost savings. Utility will be quantified in quality-adjusted life years. We will elicit health-related quality of life during the study period using two preference based measures of health status: the Child Health Utility 9D[31] and the EuroQol 5D-Y.[32] Future health and cost impacts, beyond the study period, will be estimated using numerical modelling. Incremental cost effectiveness ratios, comparing the closed-loop system to usual care will be calculated.

### Interim analysis
We will not perform an interim analysis.

### Perprotocol analysis
We will conduct a perprotocol analysis in order to replicate the primary analysis, but limited to participants who did not withdraw from the study (withdrawals excluded even if they return for a 6-month glycated haemoglobin measurement) and used closed-loop for at least 70% of the time (intervention group).

### Power calculation
Data from the JDRF Continuous Glucose Monitoring Randomised Clinical Trial (JDRF CGM RCT)[33] from subjects who would have met the eligibility criteria for the current trial were used to project the distribution of baseline and 6-month glycated haemoglobin. Among n=53 subjects meeting the eligibility criteria in the JDRF CGM RCT (n=20 subjects 8–12 years of age and n=33 subjects 13–18 years of age), the upper limit of the CI for the effective SD of glycated haemoglobin was 0.71%. With this effective SD, for a true 0.4% reduction in glycated haemoglobin, power=85%, two-sided type 1 error=5%, 1:1 randomisation, total sample size is estimated to be 116. Adding 10% for potential dropout/non-compliance

results in a final total sample size of ~128 (64 in each treatment group).

## STUDY MANAGEMENT

### Data Safety Monitoring Board

A Data Safety Monitoring Board (DSMB) will be instituted. The DSMB will be notified of all serious adverse events and any unanticipated adverse device effects/events and will perform regular safety data review. The DSMB will report to the National Institute of Diabetes and Digestive and Kidney Diseases (the Funder) any safety concerns and recommendations for suspension or early termination of the trial.

### Study sponsors

In the UK, the study sponsors are the University of Cambridge and the Cambridge University Hospitals NHS Foundation Trust. Study sponsor in the USA is the Jaeb Center for Health Research.

### Study management committee

A study management committee composed of the chief investigator, study coordinators and study data manager will meet monthly to discuss the operational aspects of the trial.

### Data management and monitoring

Designated personnel from coordinating centres will be responsible for maintaining quality assurance and quality control systems to ensure that the trial is conducted and data are generated, documented and reported in compliance with the protocol, Good Clinical Practice and regulatory requirements.

We will observe confidentiality of subject data. Personal details for each participant with a link to a unique identification number will be held locally on a study screening log in the Trial Master File at each of the investigation centres. These details will not be disclosed at any other stage during the study, and all individual results will remain anonymous.

### Indemnity

Indemnity for any harm rising on the conduct of research will be provided according to arrangements in respective countries:
1. UK—any liability arising from study design will be covered by clinical trial insurance policy organised by the University of Cambridge. National Health Service indemnity cover will apply for any claims arising from management and conduct of research.
2. USA—any liability arising from study design will be under the responsibility of the participants or their insurance company.

## ETHICS AND DISSEMINATION

The study has undergone a review by regulatory authorities in the UK (Medicines and Healthcare products Regulatory Agency) and in the USA (Food and Drug Administration). All participants will be provided with oral and written information about the trial and procedures involved in the study before obtaining written informed consent. For minors, parents/guardians will provide written informed consent, and written assent will be gained.

Standard operating procedures for monitoring and reporting of all adverse events and adverse device effects will be in place including serious adverse events, serious adverse device effects and specific adverse events, such as severe hypoglycaemia and significant hyperglycaemia with ketosis.

Any substantial amendments to the protocol and other documents shall be submitted to, and approved by, the Independent Research Ethics Committee/Institutional Review Board and the regulatory authorities, prior to implementation as per nationally agreed guidelines.

The study started enrolling participants in June 2017 and is expected to complete clinical follow-up by January 2020 and to report results in 2020. Trial results will be disseminated in internationally peer-reviewed scientific journals.

**Author affiliations**

[1]Wellcome Trust-MRC Institute of Metabolic Science, University of Cambridge, Cambridge, UK
[2]Department of Paediatrics, University of Cambridge, Cambridge, UK
[3]Department of Diabetes and Endocrinology, Cambridge University Hospitals NHS Foundation Trust, Cambridge, UK
[4]Department of Pediatrics and Adolescent Medicine, Medical University of Vienna, Vienna, Austria
[5]Department of Paediatric Diabetes, Leeds Children's Hospital, Leeds, UK
[6]Department of Paediatric Diabetes and Endocrinology, Nottingham University Hospitals NHS Trust, Nottingham, UK
[7]Department of Paediatric Endocrinology and Diabetes, Southampton Children's Hospital, Southampton General Hospital, Southampton, UK
[8]Barbara Davis Center for Childhood Diabetes, University of Colorado, Aurora, Colorado, USA
[9]Department of Pediatrics, Division of Pediatric Endocrinology and Diabetology, Wells Center for Pediatric Research, Indiana University School of Medicine, Indianapolis, Indiana, USA
[10]Division of Pediatric Endocrinology, Stanford University, Stanford, California, USA
[11]Department of Pediatrics, Yale University, New Haven, Connecticut, USA
[12]Department of Pharmaceutical and Health Economics, School of Pharmacy, University of Southern California, Los Angeles, California, USA
[13]Jaeb Center for Health Research, Tampa, Florida, USA

**Acknowledgements** Jasdip Mangat supported development and validation of closed-loop system. Josephine Hayes, Matthew Haydock and Nicole Ashcroft (Institute of Metabolic Science, University of Cambridge) provided administrative support. NIHR Cambridge Clinical Research Facility will support the research team in their research-related activities. Artificial Pancreas focus group contributors provided feedback on the study design. West Midlands Young Persons Advisory Group reviewed Participant Information Sheets.

**Contributors** RH, MEW, FC, LD, NT, PW, LDM, BAB, SW, CLA and KH codesigned the study. CK and PC designed the statistical plan. GM, JMA, SH, MT, CB, FC, LD, NT, PW, LDM, BAB, SW and CLA screened and enrolled participants, arranged informed consent from the participants, provided patient care and took samples. KH devised the human factors assessments. JS and SB coordinated the study. JS managed randomisation. SF will conduct the cost utility analysis. RH designed and implemented the glucose controller. GM and RH wrote the manuscript. All authors critically reviewed the report. No writing assistance was provided.

**Funding**  National Institute of Diabetes and Digestive and Kidney Diseases of the National Institutes of Health under Award Number UC4DK108520. Additional support for the artificial pancreas work is from National Institute for Health Research Cambridge Biomedical Research Centre, and Wellcome Trust Strategic Award (100574/Z/12/Z). Medtronic is supplying discounted CGM devices, sensors and details of communication protocol to facilitate real-time connectivity. Abbott Diabetes Care is providing Libre Pro sensors.

**Competing interests**  RH reports having received speaker honoraria from Eli Lilly and Novo Nordisk, serving on advisory panel for Eli Lilly and Novo Nordisk, receiving licence fees from BBraun and Medtronic. RH and MEW report patent patents and patent applications. MT has received speaker honoraria from Medtronic and NovoNordisk. PW reports receiving speaker honoraria from Dexcom and serving on advisory panels for Eli Lilly and Novo Nordisk and research support from Bigfoot Biomedical, Dexcom, Lexicon, Mannkind and Novo Nordisk. BAB is on Advisory Boards for Novo Nordisk and Convatec, has received research support from Medtronic Diabetes, Tandem Diabetes, Insulet, Convatec and Dexcom. SW has received speaker honoraria from Medtronic, Insulet and Tandem, and has received consultant honoraria from Sanofi and Zealand Pharmaceuticals. KH has received research support from Dexcom for an investigator-initiated project; he has received consultant fees from Lilly Innovation Center, Bigfoot Biomedical and Insulet. LDM reports grants from Medtronic.

**Patient consent for publication**  Parental/guardian consent obtained.

**Ethics approval**  Approval from independent Research Ethics Committee/Institutional Review Board (UK, East of England-Cambridge South Research Ethics Committee, #16/EE/0380; USA, Jaeb Center for Health Research Institutional Review Board certified by the Office for Human Research Protections, FWA #00000024) has been obtained.

**Provenance and peer review**  Not commissioned; externally peer reviewed.

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
