## [Reviewer comments · BMJ Open]

ARTICLE DETAILS

TITLE (PROVISIONAL)	Assessing the efficacy, safety and utility of 6 month day-and-night automated closed-loop insulin delivery under free living conditions compared to insulin pump therapy in children and adolescents with type 1 diabetes: an open-label, multi-centre, multi-national, single-period, randomised, parallel group study protocol
AUTHORS	Musolino, Gianluca; Allen, Janet; Hartnell, Sara; Wilinska, Malgorzata; Tauschmann, Martin; Boughton, Charlotte; Campbell, Fiona; Denvir, Louise; Trevelyan, Nicola; Wadwa, Paul; DiMeglio, Linda; Buckingham, Bruce A.; Weinzimer, Stuart; Acerini, Carlo; Hood, Corey; Fox, Steven; Kollman, Craig; Sibayan, Judy; Borgman, Sarah; Cheng, Peiyao; Hovorka, Roman

VERSION 1 - REVIEW

REVIEWER	Goran Petrovski Sidra Medicine, Qatar
REVIEW RETURNED	12-Dec-2018

GENERAL COMMENTS	The submitted protocol is very interesting, which aims to evaluate the algorithm for hybrid closed loop. We have to acknowledge that it is very difficult to manage this kind of study, where different devices should be connected. The protocol is well designed, easy to read and understand. However, several comments should be addressed: 1. Abstract and methods. Please specify and clarify: insulin pump therapy is standalone therapy?2. Run in period "The run in period may be extended for blinded CGM..." If it will be extended it can influence the treatment adjustment which will generate bias. Some patients will spend more time in fine tuning comparing with others. Please correct.3. Treatment period "2. Usual care" Please specify the insulin pump as a standalone device or sensor augmented pump. Is it predefined? Please be clearer.4. Refresher training for usual care Please specify the topics for the refresher training for usual care, and if you can include time spent for both groups on education, re-education and refresher training.5. Device download Please specify the frequency and type of downloads (weekly vs daily, Carelink software or other)
---

	6. Other comments - It will add more impact to evaluate time spend in closed loop system. - Please specify preprogramed basal rates if not in closed loop (Same settings as previous)? - Stress Low glucose suspend and predictive low glucose suspends thresholds
--	---

REVIEWER	Professor Katharine Barnard Bournemouth University, UK BHR Ltd, UK
REVIEW RETURNED	13-Dec-2018

GENERAL COMMENTS	This is a well-designed clinical trial across several centers internationally.
--

REVIEWER	Nick Oliver Imperial College, London UK
REVIEW RETURNED	25-Jan-2019

GENERAL COMMENTS	Many thanks for asking me to review this protocol paper - it is well written, clear and detailed. In the strengths and limitations box the authors include some potential limitations of the study design but there is no discussion section in the main manuscript where these are expanded. I note too, that the choice of control group is not discussed: The addition of CGM, low-glucose suspend, and automated insulin delivery may be considered a complex intervention over SMBG-led pump therapy and this should be mentioned.
---

VERSION 1 – AUTHOR RESPONSE

REVIEWERS' COMMENTS:

Reviewer#1 (Goran Petrovski)

The submitted protocol is very interesting, which aims to evaluate the algorithm for hybrid closed loop. We have to acknowledge that it is very difficult to manage this kind of study, where different devices should be connected.

The protocol is well designed, easy to read and understand. However, several comments should be addressed:

1. Abstract and methods. Please specify and clarify: insulin pump therapy is standalone therapy?

RESPONSE: The control group will continue the usual care (conventional or sensor-augmented insulin pump therapy). We modified the manuscript in order to clarify this important information.

2. Run in period

“The run in period may be extended for blinded CGM...” If it will be extended it can influence the treatment adjustment which will generate bias. Some patients will spend more time in fine tuning comparing with others. Please correct.

RESPONSE: We thank the Reviewer for raising this point. The run-in will be extended if limited sensor data is available. The longer run-in will not be used for more fine-tuning as the treatment adjustments will be performed only at visit 2 for both study groups (after reviewing the sensor data). The manuscript has been revised appropriately.

3. Treatment period

“2. Usual care”

Please specify the insulin pump as a standalone device or sensor augmented pump. Is it predefined? Please be clearer.

RESPONSE: Following the suggestion raised previously this has been clarified in the manuscript (See response to point 1).

4. Refresher training for usual care

Please specify the topics for the refresher training for usual care, and if you can include time spent for both groups on education, re-education and refresher training.

RESPONSE: The topics for the refresher training for usual care are: advanced boluses, temporary basal, infusion set change, sensor calibrations. We specified them in the appropriate section of the manuscript.

The time spent on training is indicated in Table 2 and 3.

5. Device download

Please specify the frequency and type of downloads (weekly vs daily, Carelink software or other)

RESPONSE: The study pump will be downloaded via CareLink as per usual practice (every clinic visit, at least every 3 months). We added this information in the revised manuscript.

6. Other comments

- It will add more impact to evaluate time spent in closed loop system

RESPONSE: Thank you. The time spent using closed-loop will be evaluated as part of utility assessment.

- Please specify preprogrammed basal rates if not in closed loop (Same settings as previous)?

RESPONSE: Thank you. Yes, pre-programmed basal rates will apply. We clarified this issue in the safety precautions section.

- Stress Low glucose suspend and predictive low glucose suspend thresholds

RESPONSE: Low glucose suspend settings are specified in the safety precautions section. We will not be using predictive low glucose suspend.

Reviewer #2 (Katharine Barnard)

This is a well-designed clinical trial across several centers internationally.

RESPONSE: We thank the Reviewer for this supportive comment.

Reviewer #3 (Nick Oliver)

Many thanks for asking me to review this protocol paper - it is well written, clear and detailed. In the strengths and limitations box the authors include some potential limitations of the study design but there is no discussion section in the main manuscript where these are expanded.

RESPONSE: We appreciate this comment. Thank you. Unfortunately the required structure of the protocol paper does not allow a discussion section to be included. We are also limited by the word count (the paper is presently at the limit).

I note too, that the choice of control group is not discussed: The addition of CGM, low-glucose suspend, and automated insulin delivery may be considered a complex intervention over SMBG-led pump therapy and this should be mentioned.

RESPONSE: We thank the Reviewer for highlighting this point. We mentioned that our study implies a complex intervention especially if we consider the subjects who remain on pump therapy only.